# Efficient Reduction Photocatalyst of 4-Nitrophenol Based on Ag-Nanoparticles-Doped Porous ZnO Heterostructure

**DOI:** 10.3390/nano12162863

**Published:** 2022-08-19

**Authors:** Shali Lin, Xiaohu Mi, Lei Xi, Jinping Li, Lei Yan, Zhengkun Fu, Hairong Zheng

**Affiliations:** School of Physics and Information Technology, Shaanxi Normal University, Xi’an 710119, China

**Keywords:** Ag nanoparticles doped porous ZnO heterostructure, photocatalysis, localized surface plasmon resonance, synergetic effect

## Abstract

Oxide-supported Ag nanoparticles have been widely reported as a good approach to improve the stability and reduce the cost of photocatalysts. In this work, a Ag-nanoparticles-doped porous ZnO photocatalyst was prepared by using metal–organic frameworks as a sacrificial precursor and the catalytic activity over 4-nitrophenol was determined. The Ag-nanoparticles-doped porous ZnO heterostructure was evaluated by UV, XRD, and FETEM, and the catalytic rate constant was calculated by the change in absorbance value at 400 nm of 4-nitrophenol. The photocatalyst with a heterogeneous structure is visible, light-responsive, and beneficial to accelerating the catalytic rate. Under visible light irradiation, the heterostructure showed excellent catalytic activity over 4-nitrophenol due to the hot electrons induced by the localized surface plasmon resonance of Ag nanoparticles. Additionally, the catalytic rates of 4 nm/30 nm Ag nanoparticles and porous/nonporous ZnO were compared. We found that the as-prepared Ag-nanoparticles-doped porous ZnO heterostructure catalyst showed enhanced catalytic performance due to the synergetic effect of Ag nanoparticles and porous ZnO. This study provides a novel heterostructure photocatalyst with potential applications in solar energy and pollutant disposal.

## 1. Introduction

Semiconductor nanomaterials have been widely explored in the field of photocatalysis because of their unique electronic and optoelectronic properties. Under light irradiation, photoexcited electrons (e^−^) and holes (h^+^) can be generated at the conduction band (CB) and valence band (VB) if the energy absorbed by the semiconductor photocatalyst is greater than the band gap. However, due to the recombination and poor optical cross-section in the visible region, only part of the photogenerated e^−^–h^+^ pairs can be transferred to the surface of the photocatalyst to drive the oxidation and redox reactions, resulting in a slow reaction rate and low catalytic efficiency [1,2,3,4]. To speed up the catalytic reaction, noble metal nanoparticles have been widely used due to their strong localized surface plasmon resonance (LSPR) in the visible and near-infrared regions [5,6,7,8,9]. Intriguingly, the integration of a semiconductor and noble metal nanoparticles is a successful example of a plasmon-enhanced photocatalyst that can inhibit e^−^–h^+^ recombination, reduce activation energy, open up new reaction pathways, lower the cost, and promote the effective utilization of solar energy [10]. The hybrid nanostructures composed of a semiconductor and plasmon metal nanoparticles are mainly designed as core–shell, noble metal nanoparticles deposited on semiconductor surfaces, Janus structure, and so on [11]. According to the contact form, they can also be divided into three types: embedded, encapsulated, and isolated forms [12]. Many approaches have been studied to add metal nanoparticles to semiconductor oxides, for example, anchor metal nanoparticles on the surface of semiconductor nanostructure by a microwave polyol process, electrodeposition, photodeposition [13,14,15], grown semiconductor shell around the preprepared metal nanocore by reducing corresponding metal ions or cation exchange, or others [16,17,18,19,20]. Compared with other plasmon nanoparticles, oxide-supported Ag nanoparticles (Ag NPs) have been widely studied as a high-performance catalyst because of being easier to synthesize and having a stronger LSPR in the visible region [21,22].

Metal–organic frameworks (MOFs) are crystalline materials formed by extensive coordination of metal ions or clusters with organic linkers, characterized by high porosity, large specific surface area, and adjustable pore size. Many kinds of nanoparticles have been encapsulated in the cavities of MOFs [23,24]. Because of the instability of the organic ligand, MOFs are used as a sacrificial precursor to obtain corresponding metal oxides by thermal decomposition; fortunately, the porous structure can still be retained [25].

As one of the water pollutants, 4-nitrophenol (4-NP) has posed a serious threat to human health; therefore, it is of great importance to reduce 4-NP to 4-aminophenol (4-AP), which is less toxic [26]. One of the most popular photocatalysts for 4-NP reduction is noble metal nanoparticles doped semiconductors. The heterostructure composed of a noble metal and semiconductor can not only enhance light absorption but also increase the junction interface, which is beneficial to electron transfer and accelerates the catalytic rate. In the presence of light irradiation, the electron is transferred between metal nanoparticles and semiconductors, so that the catalytic reactions are accelerated. However, the role of the hot electrons excited by the LSPR of noble nanoparticles in a heterostructure was ignored in most studies [27,28,29,30,31].

In this work, a new Ag NPs doped porous ZnO (Ag/p-ZnO) heterostructure material that responds to visible light was successfully synthesized by calcining Ag^+^-doped MOFs. The catalytic activity and mechanism of the heterostructure under dark, UV, and visible light were analyzed. Four mixing kinds of 4 nm/30 nm Ag NPs and porous and nonporous-ZnO (p and non-p-ZnO, respectively) over 4-NP were compared. The results showed that in addition to the synergistic effect of the heterostructure, the LSPR of Ag NPs excited by visible light also has a strong influence on the electron transfer efficiency from NaBH_4_ to 4-NP. These findings help to understand the importance of plasmon catalysis from noble nanoparticles in the heterostructure and design of a photocatalyst with a reasonable structure and effective absorption of visible light.

## 2. Experimental

### 2.1. Materials

In this study, we used zinc nitrate hexahydrate (Zn(NO_3_)_2_·6H_2_O), zinc acetate (Zn(CH_3_COO)_2_), sodium hydroxide (NaOH), adenine (C_5_H_5_N_5_), 4,4’-stilbenedicarboxylicacid (C_16_H_12_O_4_), N, N-dimethylformamide (DMF), purchased from Sigma-Aldrich, St. Louis, MO, USA. Silver nitrate (AgNO_3_), sodium borohydride (NaBH_4_), 4-nitrophenol (4-NP), and hexadecyl trimethyl ammonium chloride (CTAC), purchased from Sinopharm Chemical Reagent Co., Ltd., Shanghai, China. All of the reagents and solvents were used without further purification.

### 2.2. Materials Sample Preparation

The process for preparing the Ag/p-ZnO heterostructure is shown in Figure 1a. The MOFs were prepared via a solvothermal reaction according to the previously reported method with minor modifications [32]. To obtain more uniform nanospheres, except for the reaction carried out under magnetic stirring at 85 °C for 5 h in the water bath, the other conditions remain unchanged. The Ag/p-ZnO heterostructure were prepared according to the previously reported method with minor modifications [33]. The as-prepared MOFs’ aqueous solution was soaked in AgNO_3_ solution and continuously stirred for 3 h at room temperature; in the meanwhile, Me_2_NH_2_^+^ was replaced by Ag^+^ via a simple cation exchange process as well as the strong interactions between Ag^+^ and the nitrogen atoms of the adenine linkers [34,35]. Ag^+^-doped MOF composites were collected by centrifugation, then the excess Ag^+^ on the surface of MOFs was washed away using DI water and dried at 60 °C for 12 h again. Finally, the Ag^+^ was reduced into Ag NPs, MOFs were thermally decomposed into p-ZnO at 500 °C in air for 2 h at a heating rate of 5 °C/min, and the powder of the Ag/p-ZnO composite was obtained.

First, 4 nm Ag NPs were synthesized by mixing an aqueous solution of CTAC (10 mL, 0.1 M) and AgNO_3_ (250 mL, 0.01 M); then, the solution became dark brown when NaBH_4_ (60 µL, 0.1 M) was added under rapid stirring. The 4 nm Ag NPs were obtained by continuously stirring for 2 min. We synthesized 30 nm Ag NPs according to the previously reported method [36]. The non-p-ZnO was prepared via a solvothermal reaction. The aqueous solution of Zn(CH_3_COO)_2_ (27.2 mL, 0.2 mol/L) was added to a beaker flask, then a NaOH aqueous solution (7.8 mL, 2.8 mol/L) was added drop by drop under continuous magnetic stirring for 10 min. After that, the sample was transferred into a Teflon-lined stainless-steel autoclave kept at 180 °C for 24 h, cooled to room temperature, then washed with DI water 2~3 times.

### 2.3. Structural and Optical Characterization

The structure and morphology of Ag/p-ZnO were characterized by X-ray diffraction (XRD, Bruker AXS, D8 advance, Karlsruhe, Germany ad field-emission transmission electron microscope (FETEM, J-2100, JEOL, Chiyoda, Japan); the UV–vis absorption spectrum was measured by a Perkin Elmer Lambda 950 spectrometer (Perkin Elmer, Lambda 950, Shanghai, China).

### 2.4. Catalytic Reduction of 4-NP

Typically, the catalyst was added to a mixture of 1 mL 4-NP (0.4 mM) and 1 mL NaBH_4_ (40 mM) aqueous solution and immediately shaken homogeneously, and then transferred to a cuvette. The absorption spectra were collected from 300 to 500 nm. The corresponding mass and volume are shown in Appendix A. For the blank experiment, 1 mL of 4-NP was mixed with 1 mL of NaBH_4_ without catalyst, and the Ag NPs solution was replaced by the same volume of DI water when 5.705 mL. A conventional 3 W UV and visible LED with 3 W was used as the light source.

## 3. Results and Discussion

### 3.1. X-ray Diffraction and Field Emission Transmission Electron Microscope 

The XRD patterns indicated that the Ag/p-ZnO heterostructures are highly crystalline (Figure 1b). All diffraction peaks could be attributed to hexagonal ZnO (JCPDS PDF NO.38-1451) or cubic Ag (JCPDS PDF NO.04-0783). The FETEM image of sample Ag/p-ZnO (Figure 1c) showed that the Ag NPs were embedded in p-ZnO.

### 3.2. Catalytic Performance

The catalytic performance of the prepared sample Ag/p-ZnO was studied for the reduction hydrogenation reaction of 4-NP using a NaBH_4_-assisted reducing agent in an aqueous solution, as shown in Figure 2a. In the absence of a catalyst, electrons cannot directly transfer from NaBH_4_ to 4-NP; therefore, they do not react and show a yellow-green color [37]. Once the catalyst was added, 4-NP reduced to 4-AP; at the same time, the yellow-green color gradually faded away and the intensity of 4-NP absorption spectra at 400 nm drastically decreased. As shown in Figure 2b, the Ag/p-ZnO sample showed the catalytic ability for 4-NP reduction under natural light, with digital photos of color changes in the insert.

To study the influence of light on the catalytic rate, the reaction was performed in the dark, under UV, and with visible light irradiation. The kinetic equation for the reduction can be written as Ln(C_t_/C_0_) = Ln(A_t_/A_0_)= −k_app_t, where C_t_/A_t_ and C_0_/A_0_ represent the concentration/absorbance of 4-NP at time t and t_0_, respectively; therefore, the rate constant k_app_ (min^−1^) can be obtained by the slope of Ln(C_t_/C_0_) versus the reaction time (min) [38]. In Figure 3, according to the change of the slope, it can be observed that regardless of being in the dark or under light irradiation, the Ag/p-ZnO heterostructure showed the catalytic ability for 4-NP reduction and relatively high catalytic activity under visible light irradiation. Additionally, the slope exhibits little difference whether in the dark or under UV light irradiation.

### 3.3. Catalytic Mechanism

It is thought that there are two pathways for electron transfer between Ag NPs and p-ZnO under light irradiation, as shown in Figure 4. The wavelength of the irradiated light determines which component of the heterostructure is excited. Under UV light, the electrons transfer from p-ZnO to Ag NPs, while under visible light, the electrons transfer occurs from Ag NPs to p-ZnO. Under UV light, the photons are absorbed by p-ZnO, then e^−^ and h^+^ are generated from CB and VB, respectively. The photoexcited electrons transfer from CB to Ag NPs due to the Schottky barrier at the interface; as a result, the recombination of e^−^–h^+^ pairs is effectively prevented. The Ag NPs serve as sinks and promote charge separation. Under visible light, due to the strong LSPR effect from Ag NPs, a large number of photoexcited hot electrons are produced on the surface of Ag NPs, which overcome the Schottky barrier during the decay of the LSPR and transfer from Ag NPs to CB [15,39]. Based on the above experimental results, it can be seen that visible light plays a more important role in promoting the catalytic rate than that under UV light. The results showed that the hot electrons induced by the LSPR of Ag NPs more easily transfer between Ag NPs and p-ZnO, and the catalytic rate is improved.

Without light, the proposed catalytic mechanism is as follows: both 4-NP and NaBH_4_ are adsorbed on the surface of Ag/p-ZnO, and NaBH_4_ in an aqueous solution reacts with hydroxyl-containing substances such as water, slowly releasing H_2_, which is dissociated into polar hydrogen H^δ−^ and H^δ+^ by Ag NPs. H^δ−^ forms Ag-H on the surface of Ag NPs, which can reduce -NO_2_ to -NH_2_ [40].

The effect of the content of Ag NPs in a Ag/p-ZnO heterostructure on the reduction rate was examined. The content of Ag NPs in Ag/p-ZnO heterostructure could be altered by changing the concentration of AgNO_3_, as shown by the different colors of powder in Appendix A. They were labeled as p-ZnO, Ag/p-ZnO-1 to -5. The XRD patterns are shown in Appendix A. Due to the ratio of silver to adenine, which could induce the transformation of MOFtoMOF, p-ZnO was surrounded by Ag NPs [34]. The UV–Vis absorption spectra (Appendix A) of p-ZnO showed a sharp absorption peak at about 370 nm due to excitonic absorption [41], while a broad absorption peak at 420–800 nm was due to the surface plasmon resonance (SPR) of Ag NPs with uneven size [42]. The absorption peak of p-ZnO gradually decreased with the increase in Ag NP loading. For samples Ag/p-ZnO-4 and Ag/p-ZnO-5, the Ag NPs absorption peak became a diagonal line, while the p-ZnO absorption peak gradually decreased. The most likely explanation for such changes is that Ag NPs were aggregated.

As the heterostructure showed a high catalytic rate under visible light, the following catalytic experiment was conducted under natural light. The catalytic activity of the heterostructure was optimized, as shown in Figure 5. The catalytic conversion of 4-NP over Ag/p-ZnO heterostructure under natural light was shown in Appendix A. The catalytic rate and conversion are shown in Appendix A. Sample Ag/p-ZnO-1 showed excellent catalytic activity with a rate of about 0.482 min^−1^ and conversion of 99%. However, we observed that the content of Ag NPs was not directly proportional to the reaction rate. For example, Ag/p-ZnO-4 and Ag/p-ZnO-5 samples had higher Ag NPs content, but the catalytic rate is reduced by about four to five times. Compared with the Ag/p-ZnO sample, the p-ZnO sample had a lower catalytic rate, so we concluded that a small amount of Ag NPs doping will greatly improve the reaction rate. 

In addition to the wavelength of radiation light and the content of Ag NPs, the contact form between Ag NPs and ZnO also affected the catalytic rate. Comparative experiments were conducted under natural light, as shown in Figure 6 and Appendix A. The mixture of p-ZnO and 30 nm Ag NPs showed higher catalytic activity than that of non-p-ZnO and 4 nm Ag NPs. However, despite the Ag NPs being attached to the surface of p/non-p-ZnO, their catalytic rates were not as fast as that of the Ag/p-ZnO heterostructure. p-ZnO had a large specific surface area and possessed more active sites; additionally, the embedded structure provided more contact interface between Ag NPs and p-ZnO, forming separated “islands” at the heterointerface [22]. We concluded that the excellent catalytic activity of the Ag/p-ZnO heterostructure is due to the Schottky barriers formed between Ag NPs and p-ZnO. The adequate contact surface between Ag NPs and p-ZnO not only facilitates electron transfer but also significantly enhances the catalytic performance of 4-NP.

## 4. Conclusions

In summary, a new visible-light responsive Ag/p-ZnO heterostructure photocatalyst was successfully synthesized using MOFs as the sacrificial precursor. The excellent catalytic performance for 4-NP indicates that the light-excited hot electrons from the LSPR can greatly accelerate the electron transfer from NaBH_4_ to 4-NP. The rate constant of the heterostructure over 4-NP can reach about 0.482 min^−1^ and is 40 times faster than that of p-ZnO under natural light. By comparing the catalytic activity of the heterostructure with that of four mixing kinds of 4 nm/30 nm Ag NPs and p/non-p-ZnO, we propose that the catalytic rate is affected by direct contact forms. Increasing the junction interface as much as possible between Ag NPs and p-ZnO is not only favorable for the catalytic reaction, but reduces the aggregation of Ag NPs. Our findings help with understanding the electron transfer process of heterostructure and designing a low-cost photocatalyst that is more environmentally friendly.

## Figures and Tables

**Figure 1 nanomaterials-12-02863-f001:**
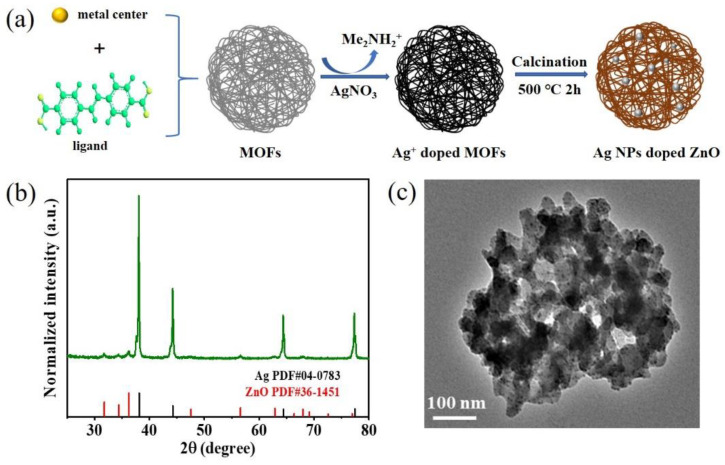
Ag-doped porous ZnO. (**a**) The synthesis procedures of Ag^+^ doped MOFs via cation exchange and Ag/p-ZnO heterostructure by calcining. (**b**) XRD patterns of Ag/p-ZnO. (**c**) TEM image of sample Ag/p-ZnO.

**Figure 2 nanomaterials-12-02863-f002:**
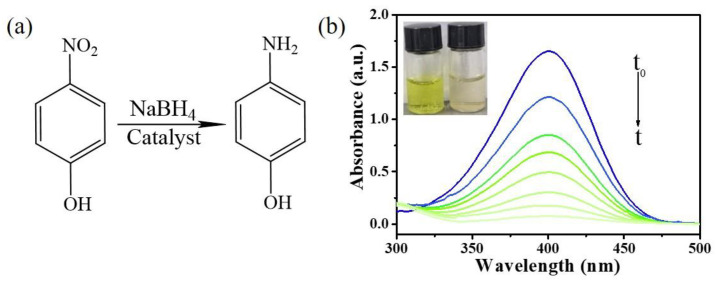
Catalytic reaction of 4-NP to 4-AP. (**a**) The catalytic reaction equation. (**b**) Time-dependent UV–Vis spectra showing gradual reduction of 4-NP over the sample Ag/p-ZnO collected in the cuvette under natural light, with digital photos of color changes before (left) and after (right) the reaction inserted.

**Figure 3 nanomaterials-12-02863-f003:**
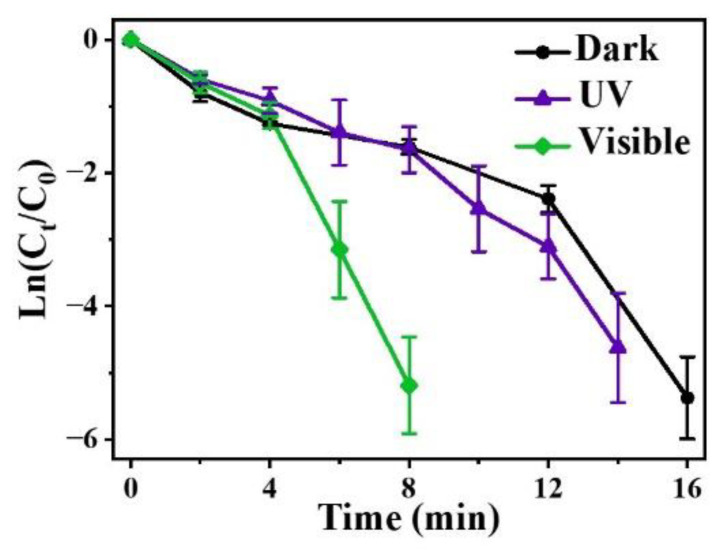
Plots of Ln(C_t_/C_0_) versus the reaction time for the reduction of 4-NP over Ag/p-ZnO in the dark, and under UV and visible light irradiation.

**Figure 4 nanomaterials-12-02863-f004:**
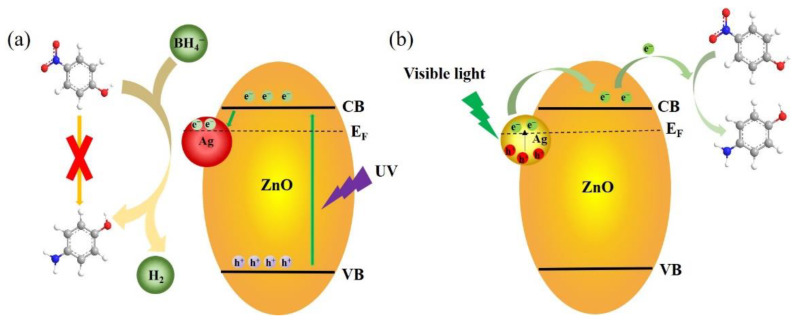
Schematic diagram of photocatalytic mechanism Ag/p-ZnO heterostructure under (**a**) UV and (**b**) visible light.

**Figure 5 nanomaterials-12-02863-f005:**
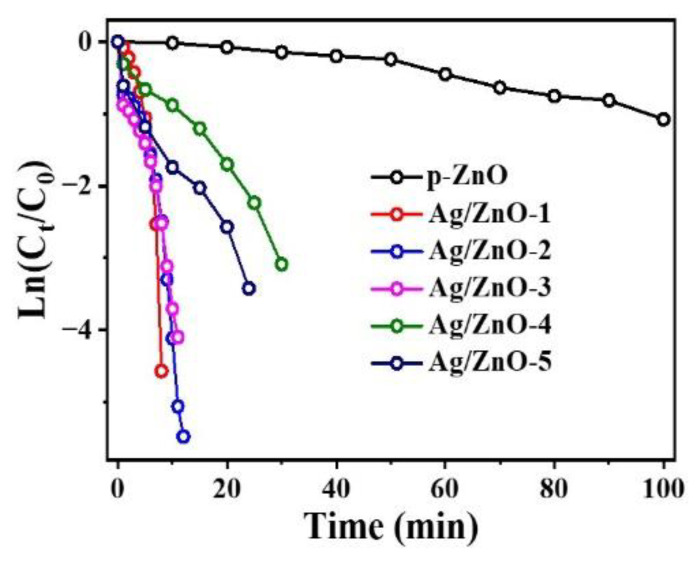
Plots of Ln(C_t_/C_0_) versus the reaction time for the reduction of 4-NP over the Ag/p-ZnO heterostructure under natural light.

**Figure 6 nanomaterials-12-02863-f006:**
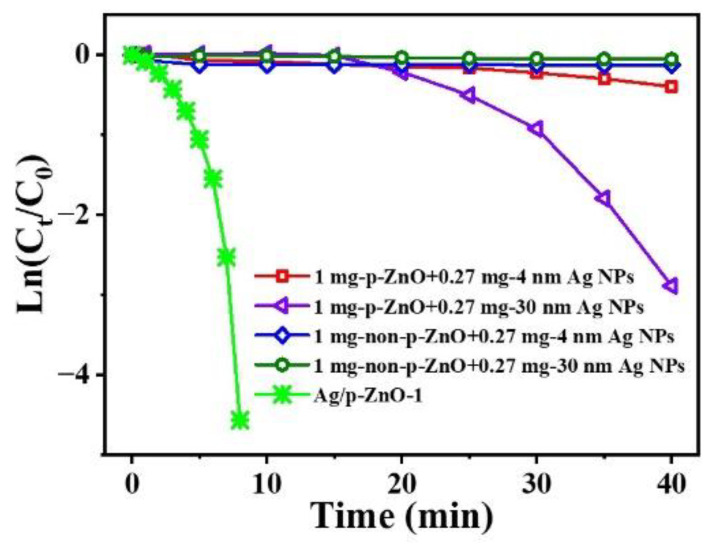
Plots of Ln(C_t_/C_0_) versus the reaction time for the reduction of 4-NP over different catalysts under natural light.

## Data Availability

The data are available on reasonable request from the corresponding author.

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
