# Peer review of "Efficient Reduction Photocatalyst of 4-Nitrophenol Based on Ag-Nanoparticles-Doped Porous ZnO Heterostructure"

_nanomaterials, 2022, doi:10.3390/nano12162863_

Round 1

Reviewer 1 Report

The work «Efficient reduction photocatalyst of 4-nitrophenol based on Ag Nanoparticles Doped Porous ZnO Heterostructure» is devoted interesting and actual for environmental protection topic. In introduction authors well describe the problem of study and substantiate the choice of object and methods used. In experimental part all materials and methods used are shown. Conclusions summarize the presented results. I have some questions and comments to improve this manuscript:

- Is it possible to estimate the band gap of synthesized photocatalysts, for example, on the basis of diffuse reflectance spectra using the Kubelka-Munk method?

- It is not entirely clear what was used as a source of UV and visible light. Were they comparable in power to compare the efficiency of photocatalysts under different conditions?

- I would like a more detailed study of the adsorption capacity of the obtained materials, since adsorption is an obligatory preliminary stage of photocatalysis (for example, using Langmuir-Hinshelwood model).

- It would be nice to compare the obtained data with the efficiency of other heterostructural photocatalysts of a similar composition or in relation to organic compounds of a similar structure.

Author Response

We thank you very much for the positive assessment and recommendation for publication. We have carefully revised the manuscript according to the reviewer’s suggestions and addressed the questions one by one.

Q1: Is it possible to estimate the band gap of synthesized photocatalysts, for example, on the basis of diffuse reflectance spectra using the Kubelka-Munk method?

Answer: The band gap of synthesized photocatalysts can be usually estimated by the Kubelka-Munk method. However, in our experimental system, the photocatalysts are Ag/p-ZnO heterostructure which is obtained by calcining Ag+-doped MOFs. As a plasmon material, Ag has a strong absorption capacity in the optical region. Therefore, the band gap measurement of the photocatalysts will be affected by the scattering and absorption of Ag, resulting in a large error.  

Q2: It is not entirely clear what was used as a source of UV and visible light. Were they comparable in power to compare the efficiency of photocatalysts under different conditions?

Answer: In the experiment, blue and white LEDs with the same power were used as light sources. They are placed at the same distance as the sample and have similar divergence angles, so the power can be considered to be similar.

Q3: I would like a more detailed study of the adsorption capacity of the obtained materials, since adsorption is an obligatory preliminary stage of photocatalysis (for example, using Langmuir-Hinshelwood model.

Answer: In general, the adsorption capacity of materials plays an important role in research work. However, our work mainly focuses on the junction interface (between Ag NPs and p-ZnO) of the catalytic reaction. According to previous studies (Sensors and Actuators B: Chemical, 2021, 334, 129667; ACS Appl. Mater. Interfaces 2019, 11, 24757; Journal of Alloys and Compounds, 2016, 658, 629;), porous ZnO indeed has higher adsorption capacity than other types of ZnO because it has more surface at the same volume. The adsorption capacity of Ag/p-ZnO heterostructure will be the focus of our next work.

Q4: It would be nice to compare the obtained data with the efficiency of other heterostructural photocatalysts of a similar composition or in relation to organic compounds of a similar structure.

Answer: In the experimental system, Ag/p-ZnO heterostructure was obtained by calcining Ag+-doped MOFs at 500 °C in the air for 2 h. Therefore, the Ag/p-ZnO heterostructure maintains the porous nature of the MOF structure. According to the reporters of the porous ZnO (International Journal of Hydrogen Energy, 2022, 47, 11190; Micro Nano Lett, 2019, 14, 1244; Ceramics International, 2017, 43, 14525;),  porous ZnO has higher catalytic efficiency than other types of ZnO. At the same time, the catalytic efficiency of two kinds of Ag/ZnO heterostructure has been shown in Figure S6. It can be seen that the Ag/p-ZnO heterostructure has a good catalytic efficiency compared with others.

Reviewer 2 Report

According to the manuscript with title: “Efficient reduction photocatalyst of 4-nitrophenol based on Ag 2 Nanoparticles Doped Porous ZnO Heterostructure ". The submitted work is introducing a new valuable and interesting idea and the given results confirm the idea. This work is suitable for publication in the Journal. I suggest the acceptance after some minor corrections as follows;

1.     Abstract section need to rewrite in correct sequence with more explanation

2.     There are some big problem in Subscript and superscript with the numbers need to be revised carefully

3.     Reformulate the aim of the work in introduction

4.     What is the size of silver and zinc oxide nanoparticles

5.     What about the recycling of the samples after the catalytic process

6.     Give some results with numbers in conclusion

7.     Discuss the new records of catalysis comparing the literature survey

8.     Add more explanation to experimental work

9.     Correct typographical errors.

10.  . You say: “The FETEM image of sample Ag/p-ZnO (Figure.1c) shows that the Ag NPs are embedded in p-ZnO with uneven sizes at the nanoscale.”, how this picture show these results although the figure don’t obtained any clear values

11.  Don’t use abbreviations in title and abstract, you must define it  in first time use

Author Response

We thank you very much for the positive assessment and recommendation for publication. We have carefully revised the manuscript according to the reviewer’s suggestions and addressed the questions one by one.

Q1. Abstract section need to rewrite in correct sequence with more explanation

Answer: The abstract section has been rewritten and more explanations have been added to the manuscript.

Q2. There are some big problems in Subscript and superscript with the numbers need to be revised carefully.

Answer: We apologize for the errors and thank the reviewer for the careful reading. The issues about the language have been thoroughly checked and corrected.

Q3. Reformulate the aim of the work in introduction

Answer: The aim of the work in the introduction has been carefully reformulated in the manuscript according to the reviewer’s suggestions.

Q4. What is the size of silver and zinc oxide nanoparticles

Answer: In the experimental system, since porous ZnO is prepared by MOFs , the size of ZnO is closely related to the size of MOFs, and the size is not uniform, ranging from several hundred nanometers to several microns. Similarly, the size of porous ZnO nanoparticles in other literatures is about 1-7 μm (Angew Chem Int Ed, 2021, 60, 6362). Because the different size of Ag nanoparticles has different absorption spectra (ACS Appl Mater Interfaces, 2019, 11, 17637; J Am Chem Soc, 2010, 132, 11372;), the size of Ag nanoparticles can be estimated by absorption spectra (Figure S3).

Q5. What about the recycling of the samples after the catalytic process

Answer: The light-excited hot electrons from LSPR can accelerate the electron transfer from BH4- to 4-NP greatly, which is the focus of our work. By comparing the catalytic activity of the heterostructure with that of four mixing kinds of 4 nm/30 nm Ag NPs and p/non-p-ZnO, it is proposed that the catalytic rate is affected by direct contact forms. Increasing the junction interface as much as possible between Ag NPs and p-ZnO is not only favorable for the catalytic reaction but reduces the aggregation of Ag NPs. The recycling of the samples will be studied in our next work.

Q6.  Give some results with numbers in conclusion

Answer: The results with numbers has been added to the conclusion of the manuscript.

Q7. Discuss the new records of catalysis comparing the literature survey

Answer: In general, it is necessary to compare new records of catalysis to the literature Survey. In this work, although great progress has been achieved regarding Ag-based catalysts for NaBH4-assisted 4-NP reduction, it is difficult to calculate exactly how much has improved. The inconsistency of catalytic performance tests, such as the different usage amounts of 4-NP, NaBH4, and catalyst, and the disunity of testing temperature, makes it difficult to fairly compare the catalytic performance of various catalysts in detail (Nano Research, 2019, 12, 2407). To prove that the catalytic performance of the synthesized sample is better, the catalytic performance of Ag NPs of different sizes, p-ZnO, and n-p-ZnO were compared under the same experimental conditions, and the results are shown in Figure. S6. It can be seen that the 30nm-Ag NPs and p-ZnO heterostructure has a good catalytic efficiency compared with others.

Q8. Add more explanation to experimental work

Answer: More explanation about experimental work has been added to the manuscript.

Q9Correct typographical errors

Answer: We apologize for the errors and thank the reviewer for the careful reading. The typographical errors in the manuscript have been thoroughly checked and corrected.

Q10. You say: “The FETEM image of sample Ag/p-ZnO (Figure.1c) shows that the Ag NPs are embedded in p-ZnO with uneven sizes at the nanoscale.”, how this picture show these results although the figure don’t obtained any clear values.

Answer: We have carefully revised the part of the manuscript according to the reviewer’s suggestions.

Q11.  Don’t use abbreviations in title and abstract, you must define it in first time use.

 Answer: The abbreviations in the title and abstract have been revised according to the reviewer’s suggestions.